# Cystic Fibrosis and Male Infertility: From Genetics to Future Perspectives in Assisted Reproductive Technologies

**DOI:** 10.3390/genes16090994

**Published:** 2025-08-25

**Authors:** Aris Kaltsas

**Affiliations:** Third Department of Urology, Attikon University Hospital, School of Medicine, National and Kapodistrian University of Athens, 12462 Athens, Greece; ares-kaltsas@hotmail.com

**Keywords:** cystic fibrosis, CFTR mutations, male infertility, CBAVD, obstructive azoospermia, assisted reproduction, sperm retrieval, ICSI, CFTR modulators, genetic counseling

## Abstract

**Background/Objectives**: Male infertility is a prevalent and often underrecognized manifestation of cystic fibrosis (CF), primarily caused by congenital bilateral absence of the vas deferens (CBAVD) due to CFTR gene mutations. With improved life expectancy in CF patients, reproductive counseling and fertility management have gained clinical relevance. **Methods**: This narrative review synthesizes current evidence on the genetic underpinnings, diagnostic evaluation, and reproductive management of male infertility in CF and CFTR-related disorders. It also highlights recent advances in assisted reproductive technologies (ART), the role of CFTR modulators, and emerging molecular research. **Results**: Most men with CF or CBAVD have intact spermatogenesis but present with obstructive azoospermia. Diagnosis relies on clinical examination, semen analysis, genetic testing, and imaging. Sperm retrieval combined with in vitro fertilization (IVF) and intracytoplasmic sperm injection (ICSI) achieves high success rates. Genetic counseling is essential to assess reproductive risks and guide partner screening. New therapies—particularly CFTR modulators—have improved systemic health and fertility potential. Future directions include gene therapy, microfluidics-based sperm selection, and personalized molecular strategies. **Conclusions**: Male infertility in CF represents a treatable consequence of a systemic disease. Advances in reproductive medicine and precision genetics now offer affected men viable paths to biological parenthood while also emphasizing the broader health implications of male infertility.

## 1. Introduction

Cystic fibrosis (CF) is a life-shortening autosomal recessive disease that affects approximately 1 in 2500–3500 live births in Caucasian populations [1]. It results from pathogenic variants in the *CFTR* gene (cystic fibrosis transmembrane conductance regulator) on chromosome 7, which encodes an epithelial ion channel crucial for fluid and electrolyte transport [2]. Dysfunctions in CFTR lead to abnormally viscid secretions in multiple organs, chiefly the lungs, pancreas, and reproductive tract [3]. While progressive pulmonary disease historically limited survival, advances in care and CFTR modulator therapies have dramatically improved life expectancy [4]. For example, the median predicted survival now exceeds four decades in many regions, and more than half of CF patients are adults rather than children [5].

Consequently, issues of fertility and family planning in CF have come to the forefront. Notably, male infertility is a cardinal extrapulmonary manifestation of CF. Up to 97–98% of men with CF are infertile due to obstructive azoospermia from congenital bilateral absence of the vas deferens (CBAVD) [6]. Nonetheless, because spermatogenesis is usually intact, many men with CF can father biological children with the aid of assisted reproductive technologies (e.g., surgical sperm retrieval and in vitro fertilization) [7].

In contrast, female reproductive capacity is relatively preserved. Most women with CF are fertile and can conceive, though thickened cervical mucus and nutritional factors may cause subfertility; with modern care, successful pregnancies in women with CF have become increasingly common [8].

This review provides a comprehensive overview of the genetic basis of CF and its inheritance, the role of CFTR mutations in male reproductive tract anomalies, the pathophysiology of CF-related male infertility, diagnostic approaches, reproductive options (including assisted reproductive technologies), clinical management and counseling, and emerging research directions such as gene therapy and personalized medicine. We also include a summary table of key CFTR mutations and their impact on male fertility. The goal is to synthesize current knowledge and recent findings to inform clinical practice and future research.

## 2. Genetic Basis of Cystic Fibrosis and Its Inheritance

CF is a multisystem disease driven by mutations in the CFTR gene. Understanding the genetic basis of CF provides insight into its diverse clinical manifestations and atypical phenotypes. This section outlines the CFTR gene and the classes of mutations that cause CF, the pattern of inheritance and carrier frequencies in different populations, and how specific CFTR genotypes correlate with disease severity and phenotypes (including effects on male fertility) [9].

### 2.1. CFTR Gene and Mutation Classes

CF results from biallelic (paired) mutations in the *CFTR* gene located on chromosome 7 (7q31.2) [10]. The gene spans 27 exons encoding a 1480 amino acid CFTR protein with multiple domains (two transmembrane domains, two nucleotide-binding domains, and a regulatory domain) that form a cyclic AMP-regulated chloride/bicarbonate channel [11]. Proper functioning of this channel is crucial for ion and fluid transport in various tissues (including the lungs, pancreas, and reproductive tract), and loss of *CFTR* function leads to the characteristic thick secretions of CF.

To date, over 2000 distinct *CFTR* variants have been identified, of which more than 700 are known to be disease-causing [12]. These mutations range from single base substitutions to large deletions, and they are traditionally categorized into six classes based on how they affect CFTR production or function [13].

In brief, CFTR mutations are classified into six functional categories based on the type of molecular defect they cause [13,14]. Class I mutations, such as nonsense or frameshift variants (e.g., G542X), result in a complete absence of functional protein due to premature termination of translation. Class II mutations, including the common F508del deletion (which removes phenylalanine at position 508), produce misfolded CFTR protein that is retained and degraded within the endoplasmic reticulum before reaching the apical membrane. Class III mutations, such as G551D, allow the CFTR protein to reach the cell surface but lead to defective gating, preventing proper channel opening and chloride transport. Class IV mutations, exemplified by R117H, impair channel conductance by reducing ion flow through the pore despite the presence of CFTR at the membrane. Class V mutations, including splice-site variants like 3849+10kb C>T or the poly-T tract 5T variant, reduce the amount of normally functioning CFTR protein by decreasing mRNA production or stability. Finally, Class VI mutations, such as C-terminal truncations like Q1412X, diminish CFTR stability at the cell surface, resulting in accelerated degradation and reduced channel half-life. These functional classes help explain the broad variability in disease severity among individuals with different CFTR genotypes.

The functional severity of a given CFTR mutation correlates with the clinical phenotype of CF. Generally, Class I–III mutations are considered “severe” mutations that usually lead to classic CF—a phenotype with multi-organ involvement (chronic pulmonary disease, pancreatic insufficiency, etc.) typically presenting in early childhood [14]. In contrast, Class IV–V mutations are often “mild” or associated with atypical CF presentations, as they allow some residual CFTR function [15].

### 2.2. Autosomal Recessive Inheritance and Carrier Frequency

CF follows an autosomal recessive inheritance pattern. An affected individual inherits one pathogenic CFTR allele from each parent. Carrier individuals are typically asymptomatic, as one functional copy of CFTR is generally sufficient for near-normal ion transport. If both parents are carriers, each child has a 25% chance of being affected by CF, a 50% chance of being a carrier, and a 25% chance of inheriting no CFTR mutation. However, if one parent has CF (i.e., two pathogenic CFTR mutations) and the other is a carrier, each child has a 50% chance of being affected by CF and a 50% chance of being a carrier, with no possibility of being completely unaffected [16]. This distinction is critical for accurate genetic counseling and reproductive planning.

The carrier frequency for CF mutations is relatively high in certain populations. In individuals of Northern European ancestry, approximately 1 in 25 (~4%) are carriers of a disease-causing CFTR mutation [17]. This high carrier rate corresponds to CF being one of the most common life-limiting recessive disorders in those populations (with an incidence of roughly 1 in 2500 newborns). Carrier frequencies and the spectrum of common mutations, however, vary by ethnicity. For example, F508del—the most prevalent CFTR mutation worldwide, accounting for around two-thirds of all CF alleles in CF patients—has a lower relative frequency in African and Asian populations [13,18]. Different groups often have their own prevalent mutations. For instance, W1282X (a nonsense mutation) is relatively common among individuals of Ashkenazi Jewish descent, and the large deletion CFTRdele2,3(21kb) (removal of exons 2 and 3) is seen at higher frequency in some Slavic and other European populations [19].

Given this genetic diversity, CF diagnostic testing and carrier screening are tailored to detect the most common mutations in the patient’s ethnic background. Standard CF mutation panels (used for newborn screening and carrier testing) typically include a set of frequent mutations found in the general population or that particular ethnic group. If a patient’s clinical presentation is strongly suggestive of CF but initial panel testing is negative, full CFTR gene sequencing is recommended to identify rare or population-specific mutations [20]. This tiered approach maximizes detection of CF-causing variants while balancing cost and efficiency.

### 2.3. Genotype-Phenotype Correlation

An individual’s *CFTR* genotype (the combination of the two mutations present) is a key determinant of their disease manifestations, though there is considerable variability even among patients with the same genotype. In general, having two “severe” CFTR mutations (Classes I–III on both alleles) usually results in classic CF, characterized by early-onset multisystem disease (chronic respiratory infections, pancreatic insufficiency, etc.) and other hallmark features like salty sweat [19]. On the other hand, milder mutation combinations such as one severe allele paired with a milder Class IV/V allele, or two mild alleles, often lead to atypical or less severe presentations. These milder phenotypes may be referred to as CFTR-related disorders (CFTR-RDs) when they do not meet the full criteria for CF [19]. CFTR-RDs represent a partial CFTR dysfunction and include conditions such as chronic sinusitis, idiopathic recurrent pancreatitis, and isolated congenital absence of the vas deferens (a cause of male infertility) without significant lung or pancreatic disease [19].

Crucially, male reproductive tract involvement can occur even in individuals without classic CF. In fact, nearly all men with classic CF have CBAVD, resulting in obstructive azoospermia and infertility. Milder CFTR genotypes can also present with reproductive issues as the primary or sole manifestation. For example, men harboring certain combinations of CFTR mutations (often one mild and one severe, such as R117H paired with a 5T splice variant) may be otherwise healthy but have CBAVD leading to infertility [21]. Notably, the intron 8 (TG)m(T)n haplotype modulates the 5T allele’s clinical impact; 5T variants with longer TG repeats (TG12 or TG13) cause more exon 9 skipping and thus confer a higher risk of CBAVD [22]. Thus, understanding CFTR genetics is crucial not only for diagnosing CF in the typical context, but also for recognizing atypical and less obvious manifestations of CFTR dysfunction such as isolated male infertility. Table 1 provides an overview of several key CFTR mutations, their functional classes, and the associated impact on male fertility.

## 3. Role of CFTR Mutations in Male Reproductive Tract Anomalies

Mutations in the CFTR gene exert a profound impact on the development and function of the male reproductive tract, particularly affecting structures derived from the Wolffian ducts. While pulmonary and gastrointestinal symptoms dominate classic CF, infertility in men, typically due to structural anomalies of the excurrent ducts, is a hallmark extrapulmonary feature [23]. This section outlines the clinical spectrum, associated genotypes, and developmental basis of CFTR-related reproductive tract anomalies.

### 3.1. Congenital Bilateral Absence of the Vas Deferens

CBAVD represents the most common male reproductive tract anomaly linked to CFTR dysfunction. It manifests as a congenital lack of the vas deferens on both sides, resulting in complete obstruction of sperm transport. In many cases, CBAVD is the only clinical feature of CFTR-related disease, particularly in men with milder or compound heterozygous genotypes [19]. Studies estimate that 95–99% of men with classic CF present with CBAVD, resulting in obstructive azoospermia [24,25]. However, the condition also occurs in men without overt pulmonary or gastrointestinal symptoms. Early molecular studies demonstrated that a substantial proportion of men with “idiopathic” CBAVD harbor one or two pathogenic CFTR variants [26].

A comprehensive meta-analysis showed that approximately 78% of men with CBAVD carry detectable CFTR mutations [27]. Of these, 46% are compound heterozygotes (biallelic), while another 28% carry a single mutation [27]. In cases with one known mutation, the poly-T tract 5T variant (a Class V mutation) often acts as a disease modifier [28]. The most frequent genotype among CBAVD patients is F508del/5T, accounting for about 17% of cases [29]. This genotype typically results in vasal agenesis while sparing other organ systems [30]. Similarly, other mild alleles, such as R117H, frequently co-occur with 5T and are enriched in the CBAVD population [31]. In contrast, men with two Class I–III mutations commonly associated with classic CF almost universally exhibit CBAVD [32].

The phenotypic spectrum of CFTR-related genital disease ranges from complete bilateral vasal agenesis in severe genotypes to normal fertility in men with only mild or polymorphic variants. As such, CBAVD lies on a continuum of CFTR dysfunction, where residual protein activity influences the degree of ductal development.

### 3.2. Other CFTR-Related Anomalies

Although CBAVD is the archetypal defect, CFTR mutations are also associated with additional structural abnormalities of the male reproductive tract. Men with congenital unilateral absence of the vas deferens (CUAVD) exhibit CFTR mutations in approximately 46% of cases—lower than in CBAVD but still significantly enriched compared to the general population [33]. Notably, the mutation profile in CUAVD often differs: while F508del and 5T variants are observed, the F508del/5T compound genotype is uncommon, suggesting that the embryologic timing or mechanism of ductal disruption may differ [33].

Beyond vasal anomalies, CFTR mutations have been implicated in idiopathic epididymal and ejaculatory duct obstruction (EDO), even when the vas deferens is present [34]. One study found that 86% of men with bilateral EDO and associated seminal vesicle abnormalities carried CFTR variants [35]. These cases underscore that CFTR dysfunction may affect multiple levels of the Wolffian-derived ductal system.

Importantly, CFTR mutations do not appear to contribute to non-obstructive azoospermia (NOA) or primary testicular failure. The prevalence of CFTR variants in men with NOA is comparable to that in the general population [36]. Although some early reports suggested that CFTR may be expressed in sperm or Sertoli cells, and potentially influence spermatogenesis, most available evidence supports a predominantly obstructive pathophysiology [37]. Men with partial ductal obstruction may demonstrate severe oligospermia—a “masked obstructive” phenotype—rather than true spermatogenic failure [37]. Thus, the male reproductive consequences of CFTR mutations are primarily attributable to anomalies of the Wolffian duct derivatives, namely, the vas deferens, epididymis, and seminal vesicles, rather than intrinsic gonadal dysfunction.

### 3.3. Developmental Pathogenesis

The mechanism by which CFTR mutations lead to CBAVD and related ductal anomalies is not fully resolved, but embryologic evidence favors a model of prenatal degeneration rather than true agenesis [32]. Two hypotheses have been proposed: (1) primary agenesis where the vas fails to form; and (2) secondary atresia where the vas initially forms but degenerates due to luminal obstruction or epithelial dysfunction in utero. The latter is better supported by histological and developmental data [32].

During embryogenesis, the Wolffian (mesonephric) duct gives rise to the vas deferens, epididymis, and seminal vesicles beginning around the seventh week of gestation, while the ureteral bud branches earlier to form the kidneys [38]. In CFTR-related anomalies, kidney development is usually preserved, with unilateral renal agenesis observed in only ~11% of CBAVD cases—much lower than in non-CFTR-related vasal anomalies [39]. This finding suggests late prenatal insult occurring after the Wolffian–ureteric separation point.

The prevailing hypothesis suggests that defective CFTR-mediated ion transport leads to thickened secretions in the developing fetal ducts. This results in luminal stasis, ductal distension, inflammation, and eventual resorption, culminating in a fibrous remnant or complete absence of the vas by birth [40]. Microscopic studies of excised vasal remnants in CBAVD patients have shown fibrotic, cord-like structures rather than a normally canalized lumen—again consistent with an acquired, in utero degeneration [41]. In men with milder mutations, intermediate phenotypes may occur, such as partial vasal formation or isolated distal epididymal obstruction.

CFTR mutations can also interfere with the development of the seminal vesicles and ejaculatory ducts. These structures arise from the distal Wolffian duct and are often hypoplastic or absent in men with CBAVD [42]. Their absence contributes to the characteristic semen profile of CF-related infertility: low volume, acidic pH, and absent fructose. In contrast, the testis and prostate derive from the genital ridge and urogenital sinus, respectively, and are typically anatomically normal in CF [43].

Lastly, an alternative genetic etiology for CBAVD must be considered in CFTR-negative cases. Mutations in the *ADGRG2* gene (also known as GPR64) cause an X-linked form of congenital vasal aplasia that mimics CBAVD but lacks systemic CF features [44]. *ADGRG2*-related infertility presents with bilateral absence of the vas, obstructive azoospermia, and intact pulmonary and pancreatic function. As this condition follows X-linked inheritance, genetic counseling implications differ markedly from autosomal recessive CFTR-related CBAVD. Identification of ADGRG2 mutations in CFTR-negative men with CBAVD is essential for accurate diagnosis, counseling, and family planning.

## 4. Pathophysiology of Male Infertility in CF Patients

Male infertility in CF is primarily due to anatomical obstruction of the reproductive tract caused by CFTR dysfunction during development. However, emerging evidence suggests that CFTR may also play more nuanced roles in sperm maturation and function beyond structural anomalies. This section addresses the dual pathophysiological mechanisms of CF-related infertility—obstructive azoospermia and potential CFTR-mediated sperm dysfunction—while highlighting their clinical and mechanistic relevance [45].

### 4.1. Obstructive Azoospermia

The hallmark of CF-related male infertility is obstructive azoospermia, defined as the complete absence of sperm in the ejaculate due to blockage or absence of the excurrent ducts. In CF and CBAVD, the obstruction is typically located at the level of the vas deferens (which is often congenitally absent) and frequently involves the epididymis as well, particularly due to atresia of its distal segments [31]. The testes themselves are usually histologically normal and capable of producing sperm, but the absence of patent ductal pathways prevents sperm from reaching the ejaculate [46].

A characteristic semen profile is observed in these patients: low ejaculate volume, acidic pH, and absent fructose, reflecting underdevelopment or agenesis of the seminal vesicles [47]. In men with incomplete ductal anomalies (e.g., partial vasal formation or focal obstruction), semen parameters may be mildly altered rather than fully diagnostic, but the underlying azoospermia still results from physical obstruction [37].

The underlying mechanism of ductal pathology has been partially elucidated through histologic studies and animal models. The CFTR protein is expressed in the epithelial lining of the male excurrent ducts, including the epididymis and vas deferens, where it regulates chloride and bicarbonate ion transport and influences fluid homeostasis. In CF, defective CFTR-mediated transport results in dehydrated, hyperviscous luminal secretions [48]. During fetal development, these thickened secretions likely obstruct the developing ductal lumen, triggering inflammation, fibrosis, and eventual degeneration, supporting the theory of prenatal atresia.

Postnatal manifestations of partial obstruction have also been described. For example, some men with CFTR mutations who have palpable vas deferens have been found to have isolated epididymal obstruction discovered during surgical exploration [49]. These individuals often present with low ejaculate volume and severe oligospermia rather than complete azoospermia [37]. Thus, CFTR dysfunction can manifest along a spectrum from complete bilateral absence of the ducts to subtle focal obstruction, leading to reduced or absent sperm output depending on severity and location. This pathophysiological cascade is illustrated in Figure 1.

### 4.2. Spermatogenesis and Endocrine Function

Despite the obstructive nature of CF-related infertility, spermatogenesis is typically preserved. Testicular biopsies in men with CF or CBAVD commonly reveal active spermatogenesis with mature sperm within the seminiferous tubules [50]. Hormonal assessments usually show normal levels of follicle-stimulating hormone (FSH), luteinizing hormone (LH), and testosterone, consistent with preserved endocrine testicular function. These findings confirm that CF does not intrinsically impair spermatogenesis in most cases.

In CF, chronic pulmonary infection and systemic inflammation generate excessive reactive oxygen species (ROS), leading to a state of oxidative stress [51]. This persistent oxidative stress can damage cellular and DNA integrity; for instance, sperm from men with CFTR mutations show increased oxidative stress-linked DNA fragmentation and epigenetic alterations, which may impair sperm quality and function [52]. Chronic illness in advanced CF can also disrupt the hypothalamic–pituitary–gonadal axis, contributing to delayed puberty, hypogonadism, or reduced libido in some patients [53]. Additionally, oxidative-stress-mediated endothelial dysfunction may lead to erectile difficulties, further complicating reproductive health [54,55]. These comorbid factors, however, are generally reversible with appropriate medical and nutritional management and do not constitute direct causes of infertility [56].

Beyond structural anomalies, emerging data suggest that CFTR may play subtle roles in sperm physiology. CFTR channels are expressed in both spermatozoa and the epididymal epithelium, where they are thought to contribute to the regulation of luminal pH, bicarbonate transport, and the ionic environment essential for sperm maturation [57]. Impaired bicarbonate-dependent activation of sperm motility has been hypothesized in CF, and altered glycerol permeability regulated by CFTR and aquaporins may also disrupt sperm osmotic balance and function [58].

Although clinical detection of such defects is limited due to the frequent presence of obstructive anomalies, rare cases of men with CFTR mutations and anatomically intact vas deferens but unexplained infertility raise the possibility of non-obstructive or subclinical functional impairment [50]. For example, a cohort study of 639 azoospermic men found that among those carrying R117H and/or F508del mutations, approximately half had palpable vas deferens yet exhibited impaired sperm retrieval outcomes—suggesting that CFTR mutations may affect spermatogenesis or sperm function independently of ductal obstruction [50]. Additionally, oxidative stress-induced epigenetic changes and altered small RNA profiles in sperm from infertile men with CFTR variants have been proposed as molecular mechanisms warranting further exploration [59,60].

Taken together, these findings support the notion that while obstructive azoospermia remains the primary mechanism of infertility in CF, a broader spectrum of CFTR-related functional effects on sperm physiology may contribute to subfertility in select cases. Ongoing research into these non-obstructive mechanisms is needed to fully elucidate the reproductive implications of CFTR dysfunction.

## 5. Diagnostic Evaluation for CF-Related Infertility

A comprehensive, stepwise diagnostic workup is recommended for males with suspected CF-related infertility. As such, assessment should combine targeted clinical examination, semen analysis, and adjunctive tests to confirm an obstructive etiology and elucidate the underlying molecular cause [61]. Importantly, endocrine evaluation (hormone levels: FSH, LH, testosterone) and scrotal/transrectal ultrasonography are advised prior to CFTR genetic testing, to verify obstructive azoospermia and rule out other causes [62]. The standard workup in these cases comprises history/physical exam, semen analysis, imaging and hormonal assessment, followed by CFTR mutation analysis, as illustrated in Figure 2.

### 5.1. History and Physical Examination

A history of chronic respiratory symptoms, pancreatic insufficiency, or a known diagnosis of CF in childhood raises immediate suspicion for CF-related infertility. However, many men with isolated CBAVD present without prior systemic symptoms. They are often referred to fertility clinics after discovery of azoospermia during infertility workup. On physical examination, the hallmark finding is bilateral absence of the vas deferens, which can be detected by careful palpation along the posterior aspect of each testis within the scrotum [63]. In CBAVD, neither part of the vas deferens is palpable as a discrete cord, and the epididymal tail may also be indurated or underdeveloped. Testicular size and consistency are typically normal, distinguishing these cases from primary testicular failure.

Importantly, CBAVD is virtually the only cause of azoospermia that can be diagnosed on physical exam alone, making clinical examination an invaluable step in evaluation. In contrast, men with CUAVD may have one non-palpable vas deferens cord and one normal side. CUAVD is sometimes discovered incidentally or during evaluation of mildly abnormal semen parameters, such as low volume or reduced motility [33].

### 5.2. Semen Analysis

The semen profile in CBAVD is distinct. Azoospermia is accompanied by very low ejaculate volume (<1 mL), acidic pH (<7.0), and absent fructose, reflecting the absence or hypoplasia of the seminal vesicles and vas deferens [47]. These features are highly suggestive of a bilateral obstructive process. In contrast, men with non-obstructive azoospermia (e.g., spermatogenic failure) generally have normal semen volume, pH, and fructose levels, since seminal vesicle function is preserved.

Basic semen parameters, when interpreted alongside physical findings, allow clinicians to differentiate obstructive from non-obstructive causes of male infertility. When the classic semen profile is combined with bilateral vasal absence, the diagnosis of CBAVD is strongly supported.

### 5.3. Genetic Testing for CFTR Mutations

Following clinical suspicion of CBAVD or CF-related infertility, genetic testing is essential to identify underlying CFTR mutations. Current guidelines from multiple professional societies including the World Health Organization, the American Society for Reproductive Medicine, and the European Association of Urology recommend CFTR mutation screening for all men with CBAVD and for any man with known CF planning biological parenthood [64,65,66].

Initial testing typically uses a panel of the most common disease-causing CFTR variants. If only one or no mutation is identified, comprehensive CFTR gene sequencing and rearrangement analysis are recommended to detect rare or population-specific mutations [67]. When a mutation is identified in the male, his female partner should also undergo CF carrier screening to assess the risk of transmitting CF to offspring. Even if the male’s mutations result only in CBAVD, combinations with a partner’s mutation could lead to classic CF in the child.

In cases where CFTR testing is negative but CBAVD is confirmed, sequencing of the ADGRG2 gene should be considered. ADGRG2 encodes a G-protein coupled receptor and is responsible for an X-linked form of vasal agenesis, typically presenting as isolated obstructive azoospermia without systemic CF symptoms [44].

Young syndrome (also known as sinusitis–infertility syndrome) is another condition to consider in the differential diagnosis. It is characterized by the triad of bronchiectasis, chronic rhinosinusitis, and obstructive azoospermia despite normal CFTR function [68]. In contrast to CF, which is caused by CFTR gene mutations, the etiology of Young syndrome is unclear (possibly related to prior environmental exposures) and CFTR genetic testing is negative. Reproductively, men with Young syndrome have an intact vas deferens but a functional obstruction of sperm transport in the epididymis due to abnormally thick secretions, resulting in azoospermia despite normal spermatogenesis [69]. This key difference explains why male infertility in Young syndrome can mimic CF clinically (both present with obstructive azoospermia and sinopulmonary issues) while arising from a distinct pathophysiology. Proper evaluation of an azoospermic male with sinopulmonary disease should therefore include CFTR mutation analysis to distinguish CF from conditions like Young syndrome [70].

### 5.4. CFTR Functional Tests

In select cases, particularly when genetic findings are inconclusive, functional assays can help confirm partial CFTR dysfunction. The classic sweat chloride test may show intermediate or mildly elevated values in men with CFTR-related infertility, reflecting reduced channel activity in epithelial tissues [71]. Additionally, nasal potential difference (NPD) testing can measure ion transport across respiratory epithelium and is useful in borderline or atypical cases [71].

These tests provide valuable evidence in men who may carry rare or uncertain CFTR variants, helping to establish the diagnosis of a CFTR-related disorder even when DNA sequencing is inconclusive.

### 5.5. Imaging and Other Evaluations

Imaging studies are useful adjuncts to confirm anatomic findings and detect associated anomalies. Scrotal ultrasonography can visualize the absence or truncation of the vas deferens and assess testicular size and architecture. Transrectal ultrasound (TRUS) may show absent or hypoplastic seminal vesicles and confirm ejaculatory duct aplasia, which are common in CBAVD [72].

Renal ultrasound is especially important, as men with vasal agenesis, particularly those without identifiable CFTR mutations, are at risk of concurrent renal anomalies such as unilateral renal agenesis or ectopic kidney [38]. Identifying a solitary kidney has clinical implications for counseling and long-term health surveillance.

Although hormone levels are typically normal in men with obstructive azoospermia, standard hormonal evaluation (FSH, LH, testosterone) should still be performed to rule out coexisting primary testicular dysfunction or endocrine abnormalities [73].

## 6. Reproductive Options and Assisted Reproductive Technologies (ART)

For men with CF or CFTR-related disorders, advances in ART have transformed biological parenthood from an unattainable goal into a realistic and often successful outcome. While infertility in this population is primarily due to obstructive azoospermia from CBAVD, spermatogenesis is typically preserved. This allows for surgical sperm retrieval followed by in vitro fertilization (IVF) with intracytoplasmic sperm injection (ICSI). The following sections outline current strategies for sperm retrieval, fertilization techniques, partner considerations, and the role of fertility preservation in adolescent and young adult males with CF.

### 6.1. Sperm Retrieval Techniques

Because spermatogenesis is intact in CF patients, viable sperm can be retrieved from the epididymis or testis, depending on anatomical findings and institutional preference. The two main approaches are (1) microsurgical epididymal sperm aspiration (MESA), a minor surgical procedure in which the epididymis is exposed under an operating microscope and fluid is aspirated from the tubules; and (2) testicular sperm extraction (TESE), which involves biopsy of testicular tissue via open or needle technique to isolate sperm [74].

In men with CBAVD, the epididymis is often at least partially developed and may contain mature sperm, making MESA the preferred first-line method in many centers. MESA provides high yields of motile sperm suitable for cryopreservation and multiple IVF cycles [75,76]. If epididymal fluid is scarce or fibrotic, TESE remains a reliable alternative. A less invasive technique, percutaneous epididymal sperm aspiration (PESA), uses a fine needle to aspirate sperm blindly through the scrotal skin; while less technically demanding, PESA generally yields lower sperm counts.

Additionally, microsurgical testicular sperm extraction (microTESE) can be employed in difficult cases. In microTESE, an operating microscope is used to systematically identify and extract seminiferous tubules most likely to contain sperm [77]. This technique is usually reserved for non-obstructive azoospermia (where sperm production is minimal) to maximize the chances of retrieval; in CF-related obstructive azoospermia, conventional TESE typically suffices since sperm are abundant throughout the testes [77].

Regardless of technique, sperm retrieval success in CF/CBAVD patients is remarkably high, routinely above 95% [78]. One series reported a 96.8% success rate using PESA in CBAVD cases [79]. This contrasts with sperm retrieval outcomes in non-obstructive azoospermia (NOA), where success is typically ≤50%. Recent studies confirm that testicular histology, endocrine parameters, and predictive nomograms enhance patient selection and TESE outcomes in this setting [77,80].

Because these retrievals are invasive, it is often recommended to cryopreserve excess sperm at the time of the procedure to avoid repeated surgery. In many cases, a single retrieval provides sufficient sperm for multiple IVF cycles [81].

### 6.2. IVF with ICSI

Once sperm are retrieved, IVF with ICSI is the standard method of achieving fertilization in cases of obstructive azoospermia. In ICSI, a single sperm is injected directly into each oocyte, bypassing the need for natural motility and penetration [82].

Couples in which the male partner has CBAVD tend to have excellent IVF/ICSI outcomes, with reported clinical pregnancy rates per cycle ranging from 50% to 75% [76]. In one large center’s experience, CBAVD couples achieved pregnancy in approximately 74% of cycles—likely reflecting the high quality of sperm and younger age of the female partners [76]. Live birth rates per cycle are slightly lower, due to miscarriage risk, but cumulative success improves with embryo cryopreservation and repeat attempts.

Importantly, there is no evidence that CFTR mutations in the sperm negatively affect embryo development or compromise IVF success. The mutation’s primary effect is anatomical (on duct formation), rather than functional (on sperm DNA integrity or fertilizing capacity) [83]. Even in theory, any subtle sperm membrane or ionic defects associated with CFTR dysfunction are effectively bypassed by the ICSI procedure.

### 6.3. Female Partner’s Health

When counseling couples affected by CF, the reproductive and general health of the female partner must also be considered. An increasing number of women with CF are reaching reproductive age and seeking to conceive, often in the context of improved health due to CFTR modulators [84]. Although female fertility in CF is often preserved, subfertility may occur due to thick cervical mucus, nutritional deficiencies, or chronic inflammation. Nevertheless, many women with CF can conceive naturally or with minimal assistance.

Pregnancy in women with CF carries potential risks including respiratory compromise, nutritional strain, and preterm birth, requiring coordinated care by a multidisciplinary team. If both partners have CF, the offspring will inherit at least one mutated allele from each parent and may be affected by CF unless preimplantation genetic testing (PGT) or donor gametes are used. These scenarios necessitate in-depth counseling with geneticists, CF care teams, and high-risk obstetricians [85].

### 6.4. Sperm Cryopreservation and Timing

Fertility preservation should be discussed early in life for male CF patients, ideally in late adolescence or young adulthood. In fact, CF care teams now initiate fertility counseling during adolescence so that patients understand infertility is an expected consequence of CF and can plan accordingly [86]. Once spermatogenesis is established, baseline hormonal and ultrasound evaluations can confirm normal testicular function, allowing early detection of any testicular abnormalities beyond the absence of the vas [87,88]. Identifying such issues promptly helps guide the optimal timing of sperm retrieval and cryopreservation, ensuring sperm are banked before any potential decline in health or fertility status.

When ejaculated sperm are absent due to CBAVD, conventional semen cryopreservation is not feasible. However, some centers offer testicular or epididymal sperm retrieval with cryopreservation for young men undergoing unrelated surgery (e.g., sinus or gastrointestinal procedures), allowing future use of surgically obtained sperm [89].

While proactive banking has theoretical benefits—particularly, in cases where health may deteriorate later in life—many men with CF remain healthy well into adulthood, enabling sperm retrieval at the time of planned reproduction. Nonetheless, early counseling ensures that patients understand their reproductive options and can access fertility services when ready [90].

Reproductive urologists and CF providers should collaborate to ensure timely referral and coordinated care. Conversations around future family planning should be introduced in a supportive, developmentally appropriate manner, allowing patients and families to make informed decisions [91].

## 7. Clinical Management and Counseling for CF Patients with Infertility

Managing infertility in men with CF or CFTR-related CBAVD requires more than surgical intervention or assisted reproduction. A comprehensive, multidisciplinary approach is essential, integrating genetic counseling, reproductive planning, psychosocial support, and proactive collaboration across specialties. The objective is not only to help patients achieve biological parenthood, but to do so in a way that considers long-term health, family dynamics, and ethical implications.

### 7.1. Genetic Counseling

Genetic counseling forms the cornerstone of care once CFTR mutations are identified in an infertile male. Men with CF or CBAVD and their partners should receive individualized counseling regarding inheritance risks, reproductive options, and implications for offspring [44]. As outlined earlier, CF is inherited in an autosomal recessive manner. If a man with CF or a CFTR-related disorder has biallelic CFTR mutations and his partner is a carrier, each pregnancy carries a 50% chance of a child affected by CF and a 50% chance of a carrier child. Importantly, all offspring will inherit at least one mutated CFTR allele, and there is no chance of an unaffected, non-carrier child in this scenario.

Reproductive technologies such as PGT and prenatal testing (e.g., amniocentesis or chorionic villus sampling) should be discussed as part of informed decision-making. For example, many couples in which one partner has CF and the other is a carrier opt for IVF with PGT to select unaffected embryos [92]. Notably, the first successful use of PGT for CF was reported in 1992, representing a milestone in reproductive genetics [93].

Even if the female partner is not a carrier, counseling remains important. All offspring in such couples will be obligate carriers, which may influence future reproductive decisions within the family. In some cases, couples may consider using donor sperm to avoid passing on mutations entirely. If a man’s health status is fragile or if IVF is deemed too burdensome, non-biological parenting options such as donor insemination, adoption, or foster parenting should be presented neutrally and supportively [94]. Before modern ART, these were often the only options available, and they remain entirely valid depending on a couple’s values and circumstances [95].

Ultimately, the goal of genetic counseling is to promote autonomy through informed choice, ensuring that couples understand all reproductive outcomes and feel empowered in their decisions [96].

### 7.2. CF Care Team Involvement

As CF survival continues to improve, reproductive health must be incorporated into routine care. CF centers are now encouraged to engage in fertility discussions early—ideally, during adolescence or young adulthood—so that patients are not caught off guard when they later wish to start a family. Informing young male patients that infertility is an expected consequence of CF enables psychological adjustment and timely planning [97].

This discussion must be approached with sensitivity, especially in adolescents, as the concept of infertility can provoke distress or denial. Many CF centers now offer structured fertility education and counseling sessions, where reproductive specialists explain available options and coordinate referrals [98].

When an adult CF patient expresses interest in fatherhood, the CF care team should collaborate closely with reproductive urologists and fertility specialists. Optimizing the patient’s health—particularly, nutritional status and pulmonary function—is critical prior to undergoing surgical sperm retrieval or initiating IVF. Conversely, in men whose health is deteriorating (e.g., approaching lung transplantation), the clinical team may recommend earlier sperm banking to preserve fertility before potential complications arise [99].

In the era of highly effective CFTR modulators, more men with CF maintain excellent health and can actively pursue family-building. Nonetheless, care must remain personalized, balancing medical readiness with emotional and logistical factors [100].

### 7.3. Addressing Health Risks and Lifestyle

Pre-conception evaluation of male CF patients should address not only fertility status but also the patient’s overall physical readiness for parenthood. Although men do not carry the pregnancy, their ability to participate in treatment and future caregiving may be influenced by their disease burden. Men with advanced lung disease may face unique challenges, such as fatigue during infancy care or infection risk to a newborn (e.g., from *Pseudomonas aeruginosa* colonization) [101].

CF specialists can provide strategies to minimize these risks, including respiratory therapy optimization, strict hygiene protocols at home, and anticipatory guidance. Comorbid conditions such as CF-related diabetes should be well-controlled, as poor glycemic regulation may impair energy levels, sexual function, and general health [102,103].

Bone density abnormalities, nutritional deficiencies, and emotional fatigue are additional considerations. While these factors are not contraindications to fatherhood, they warrant open and proactive discussion. With modern therapy, median survival now exceeds 50 years in many countries, making it increasingly realistic for men with CF to raise children and remain actively involved in family life [104]. Moreover, male infertility has been linked to broader health vulnerabilities and reduced life expectancy, underscoring the importance of comprehensive health evaluation in this population [105]. Gentle conversations around contingency planning (e.g., advance directives or backup caregiving plans) may also be appropriate, particularly in cases of severe or progressive disease.

### 7.4. Psychosocial Support

Infertility in CF can be emotionally burdensome, compounding the psychological impact of a lifelong chronic illness. Men may experience feelings of guilt, inadequacy, or grief over the inability to conceive naturally. This is often intensified when IVF is required, especially if the partner is also a CF carrier or affected by CF, as the emotional and financial burden may be greater [106].

Mental health professionals, particularly those with experience in chronic illness or reproductive counseling, should be integrated into fertility care for CF patients. Peer support programs may also provide value, especially when individuals can hear positive narratives from other men with CF who have successfully become fathers [107].

Sexual dysfunction such as erectile difficulties or reduced libido should not be overlooked. These symptoms may arise from physical causes (e.g., chronic hypoxia, endocrine abnormalities) or from psychological stress. Fortunately, treatment options are available, including phosphodiesterase type 5 (PDE5) inhibitors, which have been shown to be safe and effective in this population [108].

## 8. Emerging Research and Future Directions

Advancements in the management of CF have transformed it from a fatal childhood disease to a chronic adult condition. As survival improves, the need to address CF-related infertility grows in parallel. Recent innovations including highly effective modulator therapies, gene-editing technologies, and personalized reproductive strategies are now redefining what is possible for male fertility in CF. This section explores promising research areas that may reshape future clinical practice.

### 8.1. CFTR Modulator Therapies and Early Intervention

The introduction of CFTR modulators—small molecule drugs that correct specific molecular defects in CFTR protein—has revolutionized CF care over the past decade. Highly effective modulator therapy (HEMT) combinations, such as elexacaftor/tezacaftor/ivacaftor (Trikafta^®^), have demonstrated substantial improvements in pulmonary function, nutritional status, and quality of life for eligible patients [109,110].

A provocative question is whether early initiation of modulators might prevent or attenuate male reproductive tract anomalies. Currently, these agents are approved for infants as young as 1–2 years, long after the embryologic development of the vas deferens has occurred [109,110]. Nonetheless, the concept of fetal intervention is gaining attention. A case report described a pregnant CF carrier treated with Trikafta after fetal ultrasound revealed signs of meconium ileus; the therapy reversed intestinal findings, and the baby was born without complications [111]. While that case focused on gastrointestinal outcomes, it highlighted the theoretical feasibility of in utero CFTR correction via transplacental drug delivery.

Extrapolating to male fertility, we can imagine future scenarios where fetal treatment might preserve vasal development in CF-affected male fetuses. However, this remains speculative. To date, no evidence supports reversal of CBAVD in postnatal life. Modulators may improve prostate and seminal vesicle function, potentially increasing ejaculate volume, but they do not restore the vas deferens in those born without it [112].

Despite this, modulators have indirectly enhanced fertility outcomes, particularly in women, through improved health and normalized cervical mucus. Cases of unanticipated spontaneous pregnancies in women with CF receiving HEMT are increasingly reported. For men, modulators improve systemic health and may facilitate surgical sperm retrieval and ART planning [113].

The era of modulator-driven health stabilization opens new possibilities for fertility preservation. For example, testicular sperm extraction and cryopreservation in adolescence may be appropriate for select patients, especially if future cell- or gene-based reproductive therapies become viable [114,115,116].

Another important consideration is modulator safety during reproduction. Although data remain limited, current evidence indicates no adverse effects on male fertility or offspring conceived during paternal exposure [117]. Multicenter surveys and pharmacovigilance data have not shown increased risks of birth defects or infertility in men using modulators [118], supporting continued treatment during family planning.

### 8.2. Gene Therapy and Gene Editing

Long envisioned as a curative approach, gene therapy for CF has advanced considerably in recent years. Early efforts using viral vectors to deliver functional CFTR to airway epithelia faced challenges due to immunogenicity and limited transduction. However, the development of mRNA-based therapies and CRISPR/Cas9 gene-editing platforms has renewed enthusiasm [119].

For instance, inhaled CFTR mRNA formulations have shown partial restoration of channel function in early-phase trials. While these therapies currently target pulmonary tissues, systemic delivery could, in theory, correct CFTR dysfunction in reproductive structures if timed early enough. A future scenario may involve somatic gene therapy in neonates, or even pre-natal delivery to preserve vasal integrity before degeneration occurs [120].

Gene editing offers the possibility of germline correction. Ex vivo CRISPR-based editing of germ cells or embryos could eliminate CFTR mutations entirely, preventing disease transmission. Although technically feasible, germline editing faces major ethical, regulatory, and safety barriers. For now, PGT remains the preferred clinical approach to avoid affected offspring [121].

Beyond correction, stem-cell–based regenerative strategies are being explored. While true regeneration of the vas deferens remains out of reach, bioengineered ductal substitutes or prosthetic sperm conduits may become feasible with continued progress in tissue engineering. Though distant, these innovations are conceptually relevant in the long-term horizon of CF fertility care [122].

### 8.3. Microfluidics and Sperm Selection

Emerging technologies in reproductive biology—particularly, microfluidics and lab-on-chip platforms—aim to refine sperm selection for assisted reproduction. These systems leverage fluid dynamics and molecular cues to isolate sperm with optimal motility, morphology, or biomolecular markers [123].

One study proposed that CFTR function is linked to sperm capacitation, and selecting sperm with intact channel activity could improve fertilization outcomes. While this remains theoretical, such tools may be useful in CF patients who have functional sperm despite ductal anomalies. In compound heterozygous men (e.g., F508del/G551D), sperm carrying the milder mutation could, in theory, be selectively used for ICSI, although current IVF protocols do not genotype individual sperm [124].

Future applications of single-sperm sequencing or CFTR-targeted sperm sorting may allow precision selection of sperm with minimal mutation burden, particularly in conjunction with embryo PGT. While not yet clinical practice, such platforms exemplify how andrology and genetics are converging in the era of precision medicine [125].

### 8.4. Molecular and Personalized Medicine

Beyond CF-specific pathways, a growing body of research in male infertility is identifying molecular biomarkers that could inform treatment decisions. For example, specific microRNA expression patterns in testicular tissue have been associated with successful sperm retrieval in men with non-obstructive azoospermia [60]. Such findings may eventually guide fertility strategies in CF patients with equivocal presentations or comorbid testicular dysfunction.

In parallel, interest is growing in the reproductive microbiome, sometimes termed the “androbactome”. Altered microbial signatures in the male genital tract have been linked to oxidative stress, DNA fragmentation, and poor sperm quality. Although largely unexplored in CF, microbiome modulation through probiotics or anti-inflammatory therapies may offer a novel adjunct to ART in this population [126].

Furthermore, multi-omics profiling of testicular aging and germline function is revealing coordinated shifts in gene expression and epigenetic regulation, which could influence fertility trajectories in men with CF as they age [127]. These molecular insights may ultimately underpin future diagnostic panels or therapeutic targets.

### 8.5. Gene Modifiers and Male Fertility

Finally, variation in fertility outcomes among men with identical CFTR mutations has prompted investigation into genetic modifiers. Why some men with F508del/5T are fertile while others have CBAVD remains unclear. Polymorphisms in genes involved in Wolffian duct development, epithelial fluid regulation, or inflammation may buffer or exacerbate CFTR-related defects [128].

An illustrative example comes from pulmonary research, where variants in TGFB1 and other loci were shown to influence CF lung disease severity. Similar modifier effects likely exist in the reproductive system but remain underexplored [129]. If protective variants are identified, they may serve as targets for small-molecule mimetics or gene-environment interaction studies.

The future of CF-related infertility management will likely be shaped by advances across multiple domains ranging from early pharmacologic intervention and gene therapy to personalized ART protocols and biomolecular diagnostics. Continued research and cross-disciplinary collaboration will be essential to translate these innovations into clinical impact [130].

## 9. Conclusions

CF, once viewed primarily as a pediatric pulmonary disorder, now presents complex reproductive challenges as more affected individuals reach adulthood. In men, infertility due to CBAVD is a hallmark extrapulmonary manifestation, often occurring in the absence of classic CF symptoms. Understanding the genetic architecture of CFTR mutations including their classification, inheritance patterns, and genotype–phenotype correlations is essential for diagnosing CF-related male infertility. A multidisciplinary approach combining exam, semen analysis, genotyping, and imaging provides a reliable pathway to confirm the etiology of infertility. Spermatogenesis is typically intact in these men, and assisted reproductive technologies—most notably, testicular or epididymal sperm retrieval combined with IVF-ICSI—enable successful biological fatherhood in the vast majority of cases. Genetic counseling is critical to assess transmission risks, guide reproductive choices, and ensure informed, individualized care. Moreover, emerging computational tools are poised to enhance variant interpretation: machine learning–based prediction algorithms can help classify CFTR variants from whole-genome sequencing data and predict their impact on male fertility, aiding clinicians in decision-making and personalized patient counseling.

As the landscape of CF care continues to evolve, driven by highly effective modulator therapies and emerging genomic technologies, so too do the opportunities for preserving and restoring male fertility. Future strategies may include earlier CFTR correction, stem-cell or gene therapies, and refined molecular diagnostics to tailor fertility interventions. Importantly, infertility in CF should not be viewed in isolation but rather as a potential gateway to comprehensive men’s health screening, especially given its association with broader systemic vulnerabilities. Long-term management must extend beyond reproductive goals to encompass health maintenance, psychosocial support, and multidisciplinary coordination. With ongoing innovation and collaborative care, the outlook for men with CF seeking fatherhood is increasingly optimistic, representative of the broader shift toward personalized, holistic CF management.

## Figures and Tables

**Figure 1 genes-16-00994-f001:**
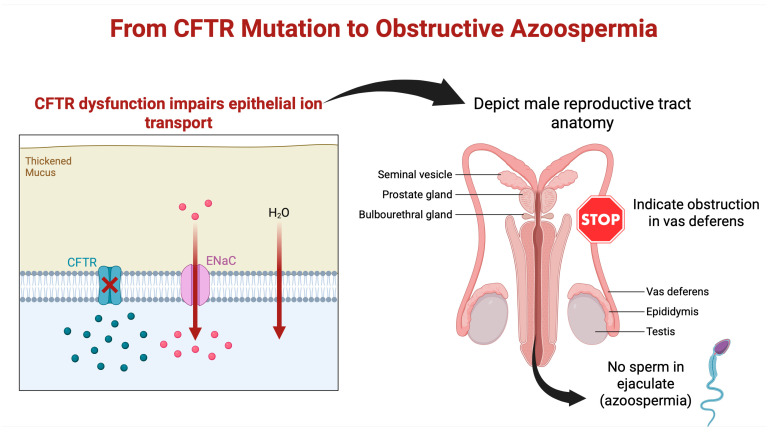
From CFTR mutation to obstructive azoospermia. CFTR mutations impair ion and water transport, resulting in thick mucus accumulation, vas deferens obstruction, and azoospermia despite normal spermatogenesis. Created in BioRender. Kaltsas, A. (2025) https://BioRender.com/q7k4dqd (accessed on 26 July 2025).

**Figure 2 genes-16-00994-f002:**
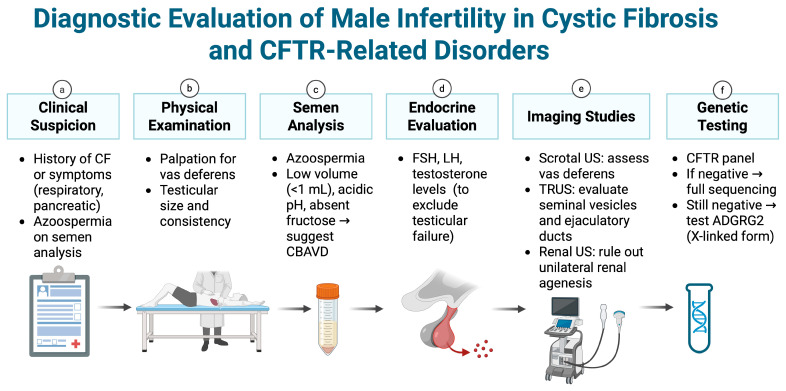
Stepwise diagnostic evaluation of male infertility in cystic fibrosis and CFTR-related disorders. Created in BioRender. Kaltsas, A. (2025) https://BioRender.com/qnwwvq9 (accessed on 19 August 2025).

**Table 1 genes-16-00994-t001:** Key CFTR variants and impact on male fertility.

CFTR Variant (Class)	Functional Effect	Clinical Severity	Male Fertility Impact
F508del (Class II)	Misfolding → little/no protein at surface	Classic severe CF (pancreatic insufficiency, lung disease)	~98% of male CF patients with F508del have CBAVD (infertility).
G542X (Class I)	Nonsense mutation → no functional protein	Classic severe CF (common in some populations)	Causes azoospermia via CBAVD in virtually all male patients (infertile).
W1282X (Class I)	Nonsense mutation → truncated protein	Severe CF (prevalent in Ashkenazi Jewish CF patients)	Associated with CBAVD in male CF (infertility expected).
N1303K (Class II)	Misprocessing of protein (trafficking defect)	Severe CF (pancreatic insufficiency)	Associated with obstructive azoospermia (CBAVD) when biallelic.
G551D (Class III)	Gating defect (channel does not open properly)	Severe CF, but responsive to ivacaftor therapy	Male patients have CBAVD and require ART; CFTR modulator therapy improves health but does not restore ducts
R117H (Class IV)	Reduced channel conductance	Variable: Often mild or atypical CF; phenotype depends on poly-T tract (see 5T)	Common in CBAVD especially when in cis with 5T variant. With 5T, can cause azoospermia even if lung/pancreas are minimally affected.
Poly-T 5T variant (Class V)	Intron 8 polymorphism reducing CFTR mRNA splicing (exon 9 skipping)	Not CF by itself; a polymorphic variant with variable penetrance	Key modifier: When one allele is 5T and the other a mild/severe mutation, commonly causes CBAVD (particularly when 5T is in cis with a TG12 or TG13 repeat). 5T alone usually does not cause infertility (seen in fertile men).
3849+10kb C>T (Class V)	Aberrant splicing with residual functional transcript	Mild CF phenotype (pancreatic sufficient; late diagnosis)	Often preserves some vas deferens function. Men with this mild mutation have a higher chance of natural fertility (reported in 2–3% of CF males), though many still have infertility.
CFTRdele2,3(21kb del) (Class I)	Deletion of exons 2–3 → no functional protein	Severe CF (second most common mutation in some regions)	Causes CBAVD in essentially all affected males (complete vas agenesis).
L138ins (p.Leu138dup, Class V)	Small in-frame insertion → partially functional protein	Mild CFTR-related disorder (common Slavic variant; often CFTR-RD)	Can contribute to CBAVD when combined with another mutation. Some men with L138ins (especially heterozygous) may be fertile or have only unilateral vas absence.

Note: CF = cystic fibrosis; CFTR = cystic fibrosis transmembrane conductance regulator; CBAVD = congenital bilateral absence of the vas deferens; ART = assisted reproductive technology, CFTR-RD = CFTR-related disorder.

## Data Availability

No new data were created or analyzed in this study.

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
