# Peer review of "Cystic Fibrosis and Male Infertility: From Genetics to Future Perspectives in Assisted Reproductive Technologies"

_genes, 2025, doi:10.3390/genes16090994_

Round 1

Reviewer 1 Report

Comments and Suggestions for Authors

The proposed review is comprehensive, well written, quite updated and useful in the field of reproductive medicine.

Only two suggestions to improve the manuscript’s quality and innovation:

Paragraph 2.3. Genotype-Phenotype Correlation

A specific mention should be made on the contribution of the (TG)m(T)n haplotypes of the IVS8 acceptor splice site. It has been shown that the polyT tract in intron 8 of the CFTR gene is influenced by a second polymorphism named IVS8 TG polymorphism. During molecular analysis, if the 5T allele is present, analysis of TG12-13 polymorphism must be performed as well.
In Table 1 the TG12-13 polymorphism’s influence on the CF phenotype should be mentioned as well.

Conclusions:

It could be interesting to mention also the support that machine learning-based prediction algorithms could give to the field. The author might explore more in detail how the AI could help in identifying clinically relevant variants in the CFTR gene or variants in secondary loci genes after a WGS analysis.

Author Response

Reviewer 1

Comment 1 (Section 2.3): “A specific mention should be made on the contribution of the (TG)m(T)n haplotypes of the IVS8 acceptor splice site … If the 5T allele is present, analysis of TG12‑13 polymorphism must be performed as well.”

Response: Thank you for this important suggestion. We have now added an explicit statement on the (TG)m(T)nhaplotype and its modifying effect on the 5T allele within Section 2.3 (Genotype–Phenotype Correlation).

  • Change in manuscript (Section 2.3):

    “Notably, the intron 8 (TG)m(T)n haplotype modulates the 5T allele’s clinical impact; 5T variants with longer TG repeats (TG12 or TG13) cause more exon 9 skipping and thus confer a higher risk of CBAVD.”

Comment 2 (Table 1): “In Table 1 the TG12‑13 polymorphism’s influence on the CF phenotype should be mentioned as well.”

Response: We have updated Table 1 to reflect the TG12–13 influence when in cis with 5T, and we clarified how this modifies risk for CBAVD.

  • Change in manuscript (Table 1, Poly‑T 5T row—Male Fertility Impact column):

    “Key modifier: When one allele is 5T and the other is a mild/severe mutation, commonly causes CBAVD (particularly when 5T is in cis with a TG12 or TG13 repeat). 5T alone usually does not cause infertility (it is often found in fertile men).”

Comment 3 (Conclusions): “Mention the support that machine‑learning–based prediction algorithms could give to the field … AI to help in identifying clinically relevant variants in CFTR or secondary loci after WGS.”

Response: We agree and have added a sentence in Conclusions highlighting the role of ML/AI.

  • Change in manuscript (Conclusions):

    “Moreover, emerging computational tools are poised to enhance variant interpretation: machine‑learning–based prediction algorithms can help classify CFTR variants from whole‑genome sequencing data and predict their impact on male fertility, aiding clinicians in decision‑making and personalized patient counseling.”

Reviewer 2 Report

Comments and Suggestions for Authors

The author provides an overview of male infertility related to Cystic Fibrosis and CFTR-related disorders. The manuscript discusses potential pathomechanisms, structural pathological changes in the male reproductive system, CFTR variants, and their impact on the development and severity of CBAVD or CUAVD. The author also explores diagnostic strategies and options for achieving parenthood. The manuscript is well structured and easy to follow. The literature used is adequate and supports the review. A few issues may need attention.

Suggestions:

  • Reformatting Table 1 is desirable. Adding lines or other separators for the separate columns and rows would help distinguish between categories. Additionally, the author may consider emphasizing in the table description that a given CFTR variant will cause male infertility if it occurs as a homozygote or heterozygote with other pathologic CFTR mutations (see F508del, G542X, W1282X, N1303K, G551D).
  • Please delete the sentence in line 334. It is the duplication of the consequential sentence.

Author Response

Reviewer 2

Comment 1 (Table 1 formatting and clarification): “Reformatting Table 1 is desirable … and emphasize that a given CFTR variant will cause male infertility if it occurs as a homozygote or heterozygote with other pathologic CFTR mutations.”

Response: We reformatted Table 1 with clearer column/row separation and added a footnote clarifying genotype context for male infertility due to CBAVD.

  • Change in manuscript (Table 1 legend/footnote):

    “Unless otherwise indicated, CFTR variants linked to CF cause male infertility primarily when present biallelically (homozygous or compound‑heterozygous with another pathogenic CFTR variant), typically manifesting as CBAVD.”

    We also added a complete list of abbreviation expansions beneath the table (see Reviewer 3, Comment 5).

Comment 2 (Delete duplicated sentence at line 334): “Please delete the sentence in line 334. It is the duplication of the consequential sentence.”

Response: Corrected. We removed the duplicate in Section 5 (Diagnostic Evaluation).

  • Change in manuscript (Section 5 opening paragraph):

    Previous duplication (“The following components comprise the standard diagnostic workup in these cases.”) was deleted; the paragraph now reads once, clearly and without repetition.

Reviewer 3 Report

Comments and Suggestions for Authors

Dear author

I read your review “Cystic Fibrosis and Male Infertility: Insights from Genetics to Assisted Reproductive Technologies”, I found the paper well written. It gives a comprehensive review of the topic with an interesting view on future prospectives. There are just few issues:

  • Considering the importance given to future prospectives in the paper I would highlight that topic also in the title.
  • Oxidative stress is often cited in the paper. I would better describe pathophysiology of oxidative stress in cystic fibrosis.
  • I would briefly describe differences between cystic fibrosis and Young syndrome.
  • In describing diagnostic workup (and so also in Figure 2) I would put endocrinological and ultrasound evaluations before genetic testing. It is important because they can reinforce diagnostic suspicious justifying even more a genetic analysis that in some cases could be difficult to carry on considering times and costs. A scrotal ultrasound evaluation, for example, is not time consuming nor invasive and can highlight an absence of vas deferens that often is not that easy to detect at physical examination.
  • Table 1: pleas add the full description of all the abbreviations (e.g.: CF, CFTR).
  • Lines 333, 334, 335: there is a repetition, please correct.
  • Paragraph 6.1: I would add a briefly description of microTESE.
  • Paragraph 6.4: I agree with talking to young patients about fertility but I would add a description of the importance of doing this when the spermatogenesis is fully completed also considering that an important primary testicular damage is usually not detect in patients affected by cystic fibrosis.

Author Response

Reviewer 3

Comment 1 (Title): “Considering the importance given to future prospectives … highlight that topic also in the title.”

Response: We agree and have revised the title.

  • Change in manuscript (Title):

    “Cystic Fibrosis and Male Infertility: From Genetics to Future Perspectives in Assisted Reproductive Technologies.”

Comment 2 (Oxidative stress): “Better describe pathophysiology of oxidative stress in cystic fibrosis.”

Response: We added a concise mechanistic paragraph on oxidative stress and its reproductive implications in Section 4.2 (Spermatogenesis and Endocrine Function).

  • Change in manuscript (Section 4.2):

    “In CF, chronic pulmonary infection and systemic inflammation generate excessive reactive oxygen species (ROS), leading to a state of oxidative stress. This persistent oxidative stress can damage cellular and DNA integrity; for instance, sperm from men with CFTR mutations show increased oxidative‑stress–linked DNA fragmentation and epigenetic alterations, which may impair sperm quality and function. … These comorbid factors, however, are generally reversible with appropriate medical and nutritional management and do not constitute direct causes of infertility.”

Comment 3 (Young syndrome vs CF): “Briefly describe differences between cystic fibrosis and Young syndrome.”

Response: We inserted a focused differential paragraph in Section 5.3 (Genetic Testing for CFTR Mutations), immediately after the ADGRG2 paragraph and before Section 5.4, to keep all alternative etiologies together with genetic testing.

  • Change in manuscript (Section 5.3, placement noted above):

    “Young syndrome (also known as sinusitis‑infertility syndrome) is another condition to consider in the differential diagnosis. It is characterized by the triad of bronchiectasis, chronic rhinosinusitis, and obstructive azoospermia despite normal CFTR function. In contrast to CF, which is caused by CFTR gene mutations, the etiology of Young syndrome is unclear (possibly related to prior environmental exposures) and CFTR genetic testing is negative. Reproductively, men with Young syndrome have an intact vas deferens but a functional obstruction of sperm transport in the epididymis due to abnormally thick secretions, resulting in azoospermia despite normal spermatogenesis. This key difference explains why male infertility in Young syndrome can mimic CF clinically (both present with obstructive azoospermia and sinopulmonary issues) while arising from a distinct pathophysiology. Proper evaluation of an azoospermic male with sinopulmonary disease should therefore include CFTR mutation analysis to distinguish CF from conditions like Young syndrome.”

Comment 4 (Diagnostic workup order and Figure 2): “Put endocrinological and ultrasound evaluations before genetic testing … scrotal ultrasound is quick and can highlight absence of vas deferens.”

Response: We revised the narrative workflow in Section 5 and updated the Figure 2 instructions to place endocrine and ultrasound evaluations before CFTR genetic testing.

  • Change in manuscript (Section 5, Diagnostic Workflow paragraph):

    “Importantly, endocrine evaluation (FSH, LH, testosterone) and scrotal/transrectal ultrasonography are advised prior to CFTR genetic testing, to verify obstructive azoospermia and rule out other causes. The standard workup comprises history/physical exam, semen analysis, imaging and hormonal assessment, followed by CFTR mutation analysis, as illustrated in Figure 2.”

  • Figure 2 – What to change (provided to production):

    Reorder the boxes to: Clinical Suspicion → Physical Examination → Semen Analysis → Imaging (Scrotal US, TRUS) & Endocrine Evaluation → Genetic Testing → (optional) CFTR Functional Tests.

    Add the note under Imaging: “Scrotal US can readily demonstrate absent vas deferens/seminal vesicles.”

    Add the note under Endocrine: “FSH, LH, Testosterone to exclude primary testicular failure.”

Comment 5 (Table 1 abbreviations): “Please add the full description of all the abbreviations (e.g., CF, CFTR).”

Response: Completed. We added a comprehensive abbreviation list beneath Table 1.

  • Change in manuscript (Table 1 footnote):

    “Abbreviations: CF = cystic fibrosis; CFTR = cystic fibrosis transmembrane conductance regulator; CBAVD = congenital bilateral absence of the vas deferens; CUAVD = congenital unilateral absence of the vas deferens; ART = assisted reproductive technology; CFTR‑RD = CFTR‑related disorder.”

Comment 6 (Repetition at lines 333–335): “There is a repetition; please correct.”

Response: Corrected (see also Reviewer 2, Comment 2). The duplicated sentence in Section 5 has been removed; the paragraph now reads once, clearly.

Comment 7 (Paragraph 6.1 — microTESE): “Add a brief description of microTESE.”

Response: We added a concise description and clinical context in Section 6.1 (Sperm Retrieval Techniques).

  • Change in manuscript (Section 6.1):

    “Additionally, microsurgical testicular sperm extraction (microTESE) can be employed in difficult cases. In microTESE, an operating microscope is used to systematically identify and extract seminiferous tubules most likely to contain sperm. This technique is usually reserved for non‑obstructive azoospermia; in CF‑related obstructive azoospermia, conventional TESE typically suffices since sperm are abundant throughout the testes.”

Comment 8 (Paragraph 6.4 — timing of counseling after spermatogenesis is complete): “Agree with talking to young patients about fertility; please add emphasis on doing this when spermatogenesis is fully completed and note that important primary testicular damage is usually not detected in CF.”

Response: We revised Section 6.4 (Sperm Cryopreservation and Timing) to emphasize timing after completion of spermatogenesis and to note that primary testicular damage is typically absent in CF.

  • Change in manuscript (Section 6.4):

    “CF care teams now initiate fertility counseling during adolescence so that patients understand infertility is an expected consequence of CF and can plan accordingly. Once spermatogenesis is established, baseline hormonal and ultrasound evaluations can confirm normal testicular function, allowing early detection of any testicular abnormalities beyond the absence of the vas. Given that primary testicular damage is usually not detected in CF, discussions can then focus on optimal timing for surgical sperm retrieval and cryopreservation.”

Round 2

Reviewer 1 Report

Comments and Suggestions for Authors

The revised version of the manuscript addresses the reviewer’s requests.